

# 21st century global glacier evolution under CMIP6 scenarios and the role of glacier-specific observations

Harry Zekollari[1,2,3,4], Matthias Huss[1,2,5], Lilian Schuster[6], Fabien Maussion[6,7], David R. Rounce[8], Rodrigo Aguayo[3], Nicolas Champollion[9], Loris Compagno[1,2,10], Romain Hugonnet[1,2,11], Ben Marzeion[12,13],
Seyedhamidreza Mojtabavi[12,14], Daniel Farinotti[1,2]

[1] Laboratory of Hydraulics, Hydrology and Glaciology (VAW), ETH Zurich, Zurich, Switzerland
[2] Swiss Federal Institute for Forest, Snow and Landscape Research (WSL), Birmensdorf, Switzerland
[3] Department of Water and Climate, Vrije Universiteit Brussel, Brussels, Belgium
[4] Laboratoire de Glaciologie, Université libre de Bruxelles, Brussels, Belgium
[5] Department of Geosciences, University of Fribourg, Fribourg, Switzerland
[6] Department of Atmospheric and Cryospheric Sciences (ACINN), Universität Innsbruck, Innsbruck, Austria
[7] Bristol Glaciology Centre, School of Geographical Sciences, University of Bristol, Bristol, UK
[8] Department of Civil and Environmental Engineering, Carnegie Mellon University, Pittsburgh, PA, USA
[9] Institut des Géosciences de l'Environnement (IGE), CNES, Grenoble, France
[10] Swiss Reinsurance Company Ltd (Swiss Re), Zürich, Switzerland
[11] University of Washington, Civil and Environmental Engineering, Seattle, WA, USA
[12] Institute of Geography, University of Bremen, Bremen, Germany
[13] MARUM – Center for Marine Environmental Sciences, University of Bremen, Bremen, Germany
[14] Center for International Development and Environmental Research, Justus Liebig University of Giessen, Giessen, Germany

*Correspondence to*: Harry Zekollari (zharry@ethz.ch, harry.zekollari@vub.be)

## Abstract

Projecting the global evolution of glaciers is crucial to quantify future sea-level rise and changes in glacier-fed rivers. Recent
intercomparison efforts have shown that a large part of the uncertainties in the projected glacier evolution is driven by the
glacier model itself and by the data used for initial conditions and calibration. Here, we quantify the effect that mass balance
observations, one of the most crucial data sources used in glacier modelling, have on glacier projections. For this, we model
the 21st century global glacier evolution under CMIP6 climate scenarios with the Global Glacier Evolution Model (GloGEM)
calibrated to match glacier-specific mass balance observations, as opposed to relying on regional mass balance observations.
We find that the differences in modelled 21st century glacier changes can be large at the scale of individual glaciers (up to
several tens of percent), but tend to average out at regional to global scales (a few percent at most). Our study thus indicates
that the added value of relying on glacier-specific observations is at the subregional and local scale, which will increasingly
allow projecting the glacier-specific evolution and local impacts for every individual glacier on Earth. To increase the ensemble
of models that project global glacier evolution under CMIP6 scenarios, simulations are also performed with the Open Global
Glacier Model (OGGM). We project the 2015-2100 global glacier loss to vary between 25±15% (GloGEM) and 29±14%
(OGGM) under SSP1-2.6 to 46±26% and 54±29% under SSP5-8.5 (ensemble median, with 95% confidence interval). Despite



some differences at the regional scale and a slightly more pronounced sensitivity to changing climatic conditions, our results agree well with the recent projections by Rounce et al. (2023), thereby projecting, for any emission scenario, a higher 21$^{st}$ century mass loss than the current community estimate from the second phase of the Glacier Model Intercomparison Project
(GlacierMIP2).

## 1 Introduction

Glaciers outside the ice sheets profoundly impact our society and the natural environment (IPCC, 2023): they act as important sea-level contributors (Marzeion et al., 2020; Edwards et al., 2021; Slangen et al., 2022), are crucial fresh water resources (Van Tiel et al., 2021; Ultee et al., 2022; Yao et al., 2022; Zanoni et al., 2023; Aguayo et al., 2023), trigger natural hazards
(Compagno et al., 2022a; Furian et al., 2022; Veh et al., 2023), regulate biodiversity (Stibal et al., 2020; Gobbi et al., 2021; Bosson et al., 2023), influence hydropower generation (Farinotti et al., 2019b; Li et al., 2022; Wasti et al., 2022), and have considerable touristic value (Purdie et al., 2020; Abrahams et al., 2022; Salim, 2023). To predict the evolution of glaciers at regional to global scales under changing climatic conditions, various types of large-scale glacier evolution models have been developed over the past two decades (for an overview see Hock et al., 2019; Marzeion et al., 2020; Zekollari et al., 2022).
Recently, these models have rapidly evolved to represent, among others, the flow of ice within glaciers (Maussion et al., 2019; Zekollari et al., 2019; Rounce et al., 2023; Bolibar et al., 2023), an advanced representation of mass balance processes (Rounce et al., 2020a; Bolibar et al., 2022; Schuster et al., 2023a), the role of (evolving) debris cover (Compagno et al., 2022b; Rounce et al., 2023; Postnikova et al., 2023), and a more realistic representation of frontal ablation and glacier calving (Huss and Hock, 2015; Recinos et al., 2021; Rounce et al., 2023; Malles et al., 2023; Recinos et al., 2023). Equally important are the many new
datasets that have been made available, which are used for model input, calibration and evaluation, including (near) global datasets on glacier outlines (RGI Consortium, 2017, 2023), ice thickness reconstructions (Farinotti et al., 2019a; Millan et al., 2022), surface velocities (Friedl et al., 2021; Millan et al., 2022), geodetic mass balances (Hugonnet et al., 2021), supraglacial debris extent and thickness (Scherler et al., 2018; Herreid and Pellicciotti, 2020; Rounce et al., 2021) and frontal ablation (Kochtitzky et al., 2022). In the second phase of the Glacier Model Intercomparison (GlacierMIP2; Marzeion et al., 2020),
these glacier evolution models and how/if they integrate the various observations were found to be the major source of uncertainty in projected glacier changes in the coming decades, as opposed to the emission scenario, the climate model providing boundary conditions, or natural variability. As such, it is of major interest to better quantify the impact that glacier evolution models and their data have on the projected glacier changes.

Projections of the global evolution of glaciers strongly depend on how glacier evolution models are calibrated to match observations (Schuster et al., 2023a). In this respect, the calibration of the mass balance component is especially important given that it determines the input and output of mass at the glacier surface (Sjursen et al., 2023; Silwal et al., 2023). Such a calibration of the mass balance component is required, since atmospheric conditions over individual glaciers cannot accurately



be represented in global-scale datasets (Aguayo et al., 2024), a problem that can only slightly be mitigated through a
downscaling and bias correction of meteorological variables (Rounce et al., 2020b). Moreover, a regional to global scale glacier
model is not able to resolve all processes that determine its mass balance, nor can it fully capture the sensitivity of each
individual glacier to climatic conditions due to microclimatic effects and/or characteristics that are not resolved in large-scale
input datasets. A detailed model calibration ensures a correct representation of the current glacier state and of its sensitivity to
changing climatic conditions. Therefore, a well-calibrated model is the prerequisite to obtain confident projections of glacier
evolution and to assess corresponding impacts.

Various approaches to calibrate the mass balance component of global glacier models have been developed. One approach
consists of calibrating the mass balance model to reproduce in-situ mass balance observations, which are available for a few
hundred glaciers worldwide (Radić and Hock, 2011; Marzeion et al., 2012; Maussion et al., 2019; Shannon et al., 2019). In
this calibration approach, initial parameters can be calibrated for glaciers that have mass balance measurements, after which
they need to be transferred to glaciers without mass balance measurements (Marzeion et al., 2012; Giesen and Oerlemans,
2013), or alternatively, regional (uniform) parameters can be calibrated to minimize the misfit with the mass balance
observations (Radić and Hock, 2011). A downside of the approach is that the modelled mass balance for the unmeasured
glaciers can be unrealistically negative or positive. Moreover, the modelled regional mass balance obtained from the
extrapolation can substantially deviate from the real regional mass balance.

A second approach calibrates the mass balance model component to match regional mass balance observations (Bliss et al.,
2014; Radić et al., 2014; Huss and Hock, 2015). This match with the regional mass balance can be obtained by assuming
model parameters to be constant for all glaciers or by varying them according to prescribed transfer functions (Bliss et al.,
2014; Radić et al., 2014). The advantage of this approach is that (physically) realistic mass balance parameters can be obtained.
However, when considering the mass balance at the individual glacier level, sometimes highly unrealistic mass balances are
obtained, resulting in an inaccurate glacier sensitivity to changing climatic conditions. Therefore, an alternative approach
which ensures a match with the regional mass balance, consists of supposing that all glaciers within a region have the same
mass balance - that is the regional mass balance (Huss and Hock, 2015). With this approach, strongly unrealistic mass balances
at the individual glacier level are prevented. However, given that in reality individual glaciers within a specific region have a
strongly varying mass balance (i.e., neighbouring glaciers can have a very different mass balance; e.g. Brun et al. (2019),
WGMS (2021)), calibrating to the regional mean can lead to substantial biases at the single-glacier level.

Until recently, the approaches presented above were considered state-of-the-art strategies when calibrating large-scale glacier
models, and the majority of the models participating in the Glacier Model Intercomparison Project (GlacierMIP; Hock et al.,
2019; Marzeion et al., 2020) relied on such approaches. A new approach now consists of calibrating the mass balance
component for every individual glacier to match glacier-specific mass balance observations. Working with such glacier-



specific (geodetic) mass balance observations allows for a leap-step in the calibration of large-scale glacier evolution models, since most of the calibration limitations described above (i.e., mismatch of regional mass balance and/or unrealistic local mass
balances) disappear. More specifically, by calibrating mass balance parameters for every glacier individually, one can match the observed mass balance for every glacier, which ensures a correct sensitivity of the glacier mass balance with respect to changing climatic conditions (assuming observations to be accurate), while also matching regional observations. Driven by the first glacier-specific mass balance observations at regional scales (Fischer et al., 2015; Brun et al., 2017; Dussaillant et al., 2019; Shean et al., 2020), a glacier-specific calibration was first used in some regional studies (Zekollari et al., 2019; Rounce
et al., 2020a; Compagno et al., 2021b, 2022b; Aguayo et al., 2023; Caro et al., 2023; Malles et al., 2023; Postnikova et al., 2023; Schuster et al., 2023a). Now, with the release of the first dataset on geodetic mass balances for every glacier on Earth (Hugonnet et al., 2021), the possibility exists to calibrate the mass balance component for every individual glacier in the world. Recently, Rounce et al. (2023) used the coupled Python Glacier Evolution Model (PyGEM, for mass balance) – Open Global Glacier Model (OGGM for ice flow and glacier evolution) model setup (hereafter referred to as "PyGEM") to project the future
global glacier evolution under various climate warming targets. This study was the first global glacier study to entirely rely on the geodetic mass balance at the individual glacier scale when calibrating the glacier-specific mass balance component. Since the work by Rounce et al. (2023) includes many novel approaches to, among others, better represent frontal ablation (glacier calving and other processes removing mass at the glacier front), debris cover, and mass balance calibration (e.g., relying on Bayesian approaches), it is not straightforward to disentangle the effect that using these new glacier-specific geodetic mass
balances has on the future modelled glacier evolution compared to the improved representation of processes.

Our study has two major objectives. First, we quantify how the data used for calibrating the mass balance model affects the projected glacier evolution. For this, we calibrate the Global Glacier Evolution Model (GloGEM; Huss and Hock, 2015) to match (i) glacier-specific and (ii) regional mass balance observations. By comparing setups with these two distinct model
calibration approaches, we isolate the effect that the mass balance calibration data has on future modelled glacier changes on various spatial scales: ranging from the glacier-specific scale, through the regional scale, to the global scale. Our second major objective is to provide new estimates on the global evolution of glaciers under CMIP6 scenarios to complement the study by Rounce et al. (2023). For this, we simulate global glacier evolution with GloGEM and the Open Global Glacier Model (OGGM; Maussion et al., 2019), with both models being calibrated at the single glacier level to match the glacier-specific geodetic mass
balance observations by Hugonnet et al. (2021). Through this effort, we aim to increase the sample of glacier models that project future glacier evolution under CMIP6 scenarios, allowing for an ensemble approach to be used when considering future glacier evolution under this latest generation of climate scenarios.



## 2 Data

### 2.1 Glacier geometry

In all simulations, glaciers are as outlined in the Randolph Glacier Inventory (RGI) version 6.0 (RGI Consortium, 2017). In GloGEM, the ice thickness is from the consensus estimate of Farinotti et al. (2019a), which is deduced from the surface elevation (as provided in Farinotti et al., 2019a) to reconstruct the bedrock elevation. In OGGM, the ice thickness is inversed assuming the Shallow Ice Approximation and mass conservation along flowlines. In the used OGGM version (v1.6.1), glacier volume is matched to Farinotti et al. (2019a) at the RGI region level, and the RGI area is matched by a dynamical spin-up (see 140 section 3.2.2).

### 2.2 Geodetic mass balance

For every glacier on Earth, a geodetic mass balance estimate is available from the global glacier elevation change dataset by Hugonnet et al. (2021). Whereas the estimates are reported at a 5-year resolution for every glacier, we here consider the trend over the full 2000-2019 period, which comes with lower uncertainties. The glacier elevation estimates of Hugonnet et al. 145 (2021) are mostly derived from time series of Advanced Spaceborne Thermal Emission and Reflection Radiometer (ASTER) digital elevation models (DEMs) (NASA/METI/AIST/Japan Spacesystems And U.S./Japan ASTER Science Team, 2001). Additionally, ArcticDEM DEMs are included when considering elevation changes of polar glaciers in the northern hemisphere (Porter et al., 2022), while Reference Elevation Model of Antarctica (REMA) DEMs are utilized for glaciers in the Antarctic periphery (Howat et al., 2019). The geodetic mass balances by Hugonnet et al. (2021) clearly illustrates that within every 150 region the spread in glacier-specific mass balances is considerable (Figure 1). This spread is particularly pronounced for small glaciers, whereas large glaciers, which dominate the (area-weighted) regional signal, are consequently closer to the regional mean. The uncertainty on these mass balance estimates partly relates to the glacier size: at the global scale, the average 1-sigma uncertainty is ~0.2 m w.e. yr$^{-1}$ for glaciers of 1 km², ~0.15 m w.e. yr$^{-1}$ for 10 km², and then goes down to around ~0.1 m w.e. yr$^{-1}$ km² for larger areas, which is the lower bound (incompressible error) due to uncertainties in density conversion 155 and uncertainties in temporal interpolation to match an exact period. The mass balance variance of individual glacier estimates is thus primarily explained by mass balance variability as opposed to their much smaller uncertainty (highlighted for RGI region 19 in Figure 1), which becomes even more relevant given that the uncertainties reported in Hugonnet et al. (2021) are slightly overestimated on average compared to validation.


**Figure 1:** Distribution of glacier-specific geodetic mass balances (MB, in m w.e. a$^{-1}$). In these plots, every point represents the 2000-2019 geodetic mass balance for an individual glacier (Hugonnet et al., 2021), whereas the red dotted line represents the region-specific (area-weighted) mean geodetic mass balance (also based on the Hugonnet et al. (2021) dataset). Every panel represents an RGI region, while the panel in lower-right corner is a zoom for the largest glaciers in RGI region 19, highlighting the relatively small uncertainties on the mass balances with respect to the mass balance variability (for the highlighted glaciers that have an area larger than 1000 km$^2$ in RGI region 19, the mean uncertainty is 0.082 m w.e. a$^{-1}$).



## 2.3 Climate forcing

In GloGEM, the past climate (until 2020) is taken from the fifth generation ECMWF reanalysis ERA5 (Hersbach et al., 2020), which combines model data with observations from across the world into a globally complete and consistent dataset. OGGM
v1.6.1 uses W5E5v2.0, which bias-adjusts ERA5 reanalysis data over land (Lange et al., 2021). For the future (from 2020 onwards), in both models (GloGEM and OGGM) the global glacier evolution is modelled under various climatic conditions that are derived from simulations performed with 12 climate models (Global Circulation Models and Earth System Models) from the Coupled Model Intercomparison Phase 6 project (CMIP6; Eyring et al., 2016): BCC-CSM2-MR, CESM2-WACCM, CESM2, EC-Earth3-Veg, EC-Earth3, FGOALS-f3-L, GFDL-ESM4, INM-CM4-8, INM-CM5-0, MPI-ESM1-2-HR, MRI-
ESM2-0, NorESM2-MM. All these climate models are run for four Shared Socio-economic Pathways (SSPs; Meinshausen et al., 2020): SSP1-2.6, SSP2-4.5, SSP3-7.0 and SSP5-8.5 (Table S 1). The EC-Earth3-Veg, GFDL-ESM4, and MRI-ESM2 climate models are also forced under SSP1-1.9 (Table S 1) but given the lower number of members (n=3), the results are not deemed representative for this emission scenario, and are therefore not discussed in this study (the glacier projections under SSP1-1.9 are however provided; see 'Data availability' section). To ensure consistency between the observational/past and the
future climate model data, a debiasing procedure is applied over the common 2000-2019 time period. For SSP1-2.6 to SSP5-8.5, these climate models are the same as used in Rounce et al. (2023), allowing for a comparison of the modelled future glacier evolution under these climate scenarios.

## 3 Methods

### 3.1 Mass balance calibration strategy

Here, we enforce the modelled specific glacier-wide mass balance ($\Delta M_{g,mod}$ modelled for each glacier $g$, in meter water equivalent (m w.e.)) to match the observed glacier-specific geodetic mass balance ($\Delta M_{g,obs}$) by Hugonnet et al. (2021):

$$\Delta M_{g,mod} = \Delta M_{g,obs} \qquad \text{(Eq. 1)}$$

Through specifically calibrated model parameters (see section 3.2), the actual mass balance of each individual glacier is thus captured, and not just the mass balance of all glaciers aggregated over an entire region. This method has been used in recent
studies with the GloGEM architecture (Zekollari et al., 2019, 2020; Compagno et al., 2021a, b, 2022a, b; Postnikova et al., 2023), OGGM (Malles et al., 2023; Aguayo et al., 2023; Caro et al., 2023; Schuster et al., 2023a), and PyGEM (Rounce et al., 2020a, 2023). Alternatively, we also evaluate how calibrating the mass balance model for every glacier to match the regional mass balance ('regional approach', commonly used until recently) affects the modelled future glacier evolution. In this regional approach, each individual glacier's specific mass balance is calibrated to match the mean regional specific mass balance during
the same multi-year time period:

$$\Delta M_{g,mod} = \Delta M_{reg,obs} \qquad \text{(Eq. 2)}$$



Here, $\Delta M_{\text{reg,obs}}$ is the observed regional mass balance in m w.e. a⁻¹ derived from the glacier-specific observations by Hugonnet et al. (2021). Hence, to match the regional mass balance, each glacier has a unique set of calibrated model parameters.

## 3.2 Specific setup for GloGEM and OGGM

### 3.2.1 GloGEM

In GloGEM (Huss and Hock, 2015), the mass balance is the sum of the (i) ablation, calculated with a degree-day model, (ii) accumulation, calculated from the precipitation and a temperature threshold to account for precipitation type, and (iii) refreezing, calculated from modelled snow and firn temperatures. The mass balance is updated annually while accounting for the evolving glacier geometry and climatic conditions. Based on this calculated mass balance, the volume change of the glacier is determined, which is then applied through elevation change hypsometric profiles that rely on observations that indicate that retreating glaciers mostly lose mass at low elevations and are relatively stable at higher elevations (referred to as "retreat parameterization" or "delta-H parameterization"; Huss et al., 2010). The mass balance calibration in GloGEM relies on a three-step calibration procedure (Huss and Hock, 2015) where, for every individual glacier, the aim is to optimally constrain (i) a multiplicative precipitation parameter ($c_{\text{prec}}$), allowing adjusting the precipitation from the climate dataset, (ii) two melt parameters, $f_{\text{snow}}$ and $f_{\text{ice}}$, relating local air temperatures and monthly melt rates over snow and ice surfaces, respectively, and (iii) a local temperature correction ($\Delta T$). The model is run over the calibration period 2000-2019 with initial estimates for the parameters $c_{\text{prec}}$ = 1.0-2.0 (region-specific, Table S 2), $f_{\text{snow}}$ = 3 mm d⁻¹ K⁻¹, $f_{\text{ice}}$ = 6 mm d⁻¹ K⁻¹ (based on literature values; Hock, 2003; Braithwaite, 2008), $\Delta T$ = 0°C. The glacier geometry is kept constant during the calibration period, and frontal mass loss for marine-terminating glaciers is computed based on this stable geometry (see Huss and Hock, 2015, for details). If the modelled glacier-wide specific mass balance of a glacier agrees with the observed geodetic mass balance (Eq. 1) within an arbitrarily set threshold of 0.01 to 0.05 m w.e. a⁻¹ to ensure convergence, the meteorological forcing series is considered to describe the climatic conditions for this glacier well, and no further changes to the parameter values are applied. If deviations are greater, $c_{\text{prec}}$ is varied between region-specific boundaries (see Table S 2) to minimize the misfit with the glacier-specific observations, until agreement is achieved (calibration step 1). The bias in precipitation is chosen as primary calibration parameter as it is expected to be most poorly captured by the climate re-analysis data and to show large small-scale variability among nearby glaciers, e.g., due to wind drift, orographic effects, and/or avalanches. If no agreement is found within the tested range, $c_{\text{prec}}$ is set to the value that result in the smallest deviation from $\Delta M_{\text{obs}}$, and $f_{\text{snow}}$ is varied, while $f_{\text{ice}}$ is adjusted so that the ratio $f_{\text{ice}}/f_{\text{snow}}$=2 is preserved (calibration step 2). If the target mass balance cannot be reproduced within these parameter ranges, we assume that there is a systematic error in the temperature forcing over the glacier. Thus, in a final step, we systematically shift the air temperature series by $\Delta T$ until agreement between the glacier's specific mass balance and the observed value is achieved (calibration step 3). Finally, in addition to the original calibration scheme as proposed in Huss and Hock (2015), this three-step calibration procedure is repeated iteratively by averaging the required shifts in the air temperature series ($\Delta T$) over every re-analysis grid cell and imposing these averages a priori before restarting the 3-step calibration cycle.





This addition aims at reducing consistent strong positive/negative temperature offsets for some re-analysis grid cells, thereby
shifting $c_{prec}$ and $f_{snow}/f_{ice}$ away from their (unrealistic) extreme bounds.

### 3.2.2 OGGM

In OGGM (Maussion et al., 2019), glaciers are represented through a flowline approach. Here we rely on OGGM v1.6.1. The
glacier evolution is calculated by solving the continuity equation for ice thickness at every point along the glacier flowline,
which accounts for the local mass balance and ice flow processes. In the OGGM setup that we use here, the glacier is
represented along one central flowline that follows the elevation bands (in a fashion similar to GloGEMflow; Zekollari et al.,
2019).

OGGM offers the possibility to calculate the mass balance in various ways (Maussion et al., 2019; Schuster et al., 2023a).
Here, we rely on OGGM's standard method that describes the mass balance through an extended version of the temperature-
index model presented by Marzeion et al. (2012). In this approach, the monthly mass balance at a given elevation is calculated
from the monthly solid precipitation and temperature, where the latter is linked to the mass balance through a temperature
sensitivity factor (μ*), which needs to be calibrated. Like GloGEM (and PyGEM; Rounce et al., 2020b), OGGM v1.6.1 also
calibrates two additional parameters at the glacier level: a multiplicative precipitation factor and a temperature bias correction
(equivalent to GloGEM's $c_{prec}$ and $\Delta T$), ensuring that the model matches observations while maintaining parameters within a
physically plausible range (Rounce et al., 2020b). The multiplicative precipitation factor is derived from an empirical
relationship between total winter precipitation and an "optimal" precipitation factor, calibrated across 114 glaciers with in-situ
winter mass balance observations (Schuster et al., 2023a, Fig. S2), applying smaller corrections for glaciers in wetter climates.
The temperature bias is determined at the climate grid point level, by fixing the melt factor to a physically reasonable value of
5 mm d$^{-1}$ K$^{-1}$ and allowing only the temperature bias to vary for calibration. The temperature bias is then fixed to the median
value of all calibrated values within that grid point. Like Rounce et al. (2020b), this calibration method results in a spatially
coherent field of temperature bias across neighbouring grid points, indicating that the temperature correction is necessary and
not purely random. Building on these initial estimates for temperature and precipitation corrections, μ* is then calibrated at
the glacier level in a three-step process similar to GloGEM, but with parameters varying within a tighter range around the
initial estimates. After this initial calibration with fixed glacier geometry, the OGGM workflow ensures that glacier mass
balance during the 2000-2020 simulation still matches observations taking elevation feedbacks into account, by recalibrating
μ* iteratively during a dynamical spin-up until observations are matched within 20% of the error estimate provided by
Hugonnet et al. (2021). This calibration process is detailed further in Aguayo et al. (2023) and in the OGGM online
documentation.



## 4 Results and discussion

### 4.1 Effect of glacier-specific mass balance calibration on future glacier evolution

We start by analysing how the future glacier evolution is affected when calibrating GloGEM's mass balance component for every individual glacier to match the glacier-specific mass balance observation (hereafter termed 'glacier-specific calibration') instead of using a regional mass balance for model calibration (hereafter 'regional calibration'; widely used approach in recent past).

### 4.1.1 Glacier scale

At the individual glacier scale, we find that differences in the modelled future glacier evolution can be substantial, particularly when the glacier-specific mass balance strongly deviates from the regional mean specific mass balance. In general, if for a given glacier the mass balance is lower than the regional one, the mass balance model parameters are calibrated to produce a more negative present-day mass balance, which translates into a more negative future mass balance and thus more substantial
projected ice loss, and vice versa for higher mass balance. This is for instance clear when considering the evolution of glaciers in the European Alps (RGI region 11 'Central Europe', Figure 2), where many of the large glaciers tend to have a mass balance that is more negative than the regional one (-0.87 m w.e. a$^{-1}$), resulting in stronger mass losses for the glacier-specific calibration. A clear example is Unteraargletscher (Switzerland; RGIv6.0 ID 11-01328; upper row in Figure 2), which – when calibrated to its strongly negative mass balance of -1.59 m w.e. a$^{-1}$ – results in a 2015-2100 volume change of -94% under the
SSP1-2.6 scenario (n=12; multi-climate model median), while this change amounts to only -73% for the regional calibration. These differences directly relate to the calibrated mass balance parameters: whereas $\Delta T$ is very similar (0.43°C vs. 0.50°C, glacier-specific vs. regional calibration, respectively), the glacier-specific calibration results in less precipitation ($c_{prec}$ of 1.30 vs. 1.57) and more melt ($f_{snow}$ of 3.15 vs. 3.00) compared to the regional calibration (Figure 3). Under a high-emission scenario (SSP5-8.5), Unteraargletscher vanishes by 2100 for both calibration cases (volume -100% and -99% vs. 2015 for glacier-
specific and regional calibration, respectively), but here the calibration approach has an important effect on the 21$^{st}$ transient evolution towards this deglaciation: e.g., the 2015-2050 volume change is -61% for the glacier-specific calibration, while the regional calibration results in a -42% change over the same time period. For Central Europe (RGI region 11), we model that for 17-25% of all glaciers (with volume >0.1km$^3$) the 2015-2050 volume projections differ by more than 10% depending on the calibration approach (Table S 3). For most other regions, these differences are even more outspoken, with for instance in
High-Mountain Asia (RGI regions 13, 14, 15), between 35-55% of all glaciers (with volume >0.1km$^3$) having differences in the 2015-2050 volume projections of more than 10% depending on the calibration approach (Table S 3). When considering the 2015-2100 volume evolution, the differences resulting from the calibration approaches are generally smaller, since a lot of the regions lose a large part of their mass, evolving to a similar (almost ice-free) state independent (Table S 3).



These differences in transient evolution directly affect projected glacier change impacts, such as those related to glacier water
discharge (Figure 3; calculated following the method as presented in Huss and Hock (2018)). For Unteraargletscher for
instance, with the glacier-specific calibration under SSP5-8.5, the annual discharge increases and peaks at values that are 15%
higher, relative to 2015, in 2050. In contrast, with the regional calibration, Unteraargletscher's annual discharge increases in
the coming decades, peaking at levels 25% higher than present in 2065. Interestingly, in the first decades (until ca. 2035-2040),

the differences in discharge between the calibration approaches are relatively limited, since the lower precipitation for the
glacier-specific calibration ($c_{prec}$ of 1.30 vs. 1.57) is in part compensated by the higher melt ($f_{snow}$ of 3.15 vs. 3.00). However,
as the glacier melts and shrinks, the effect of the melt parameter reduces, and the difference in $c_{prec}$ determines the differences
in precipitation and thus discharge (since discharge is calculated over initial glacier area) (Schuster et al., 2023a).

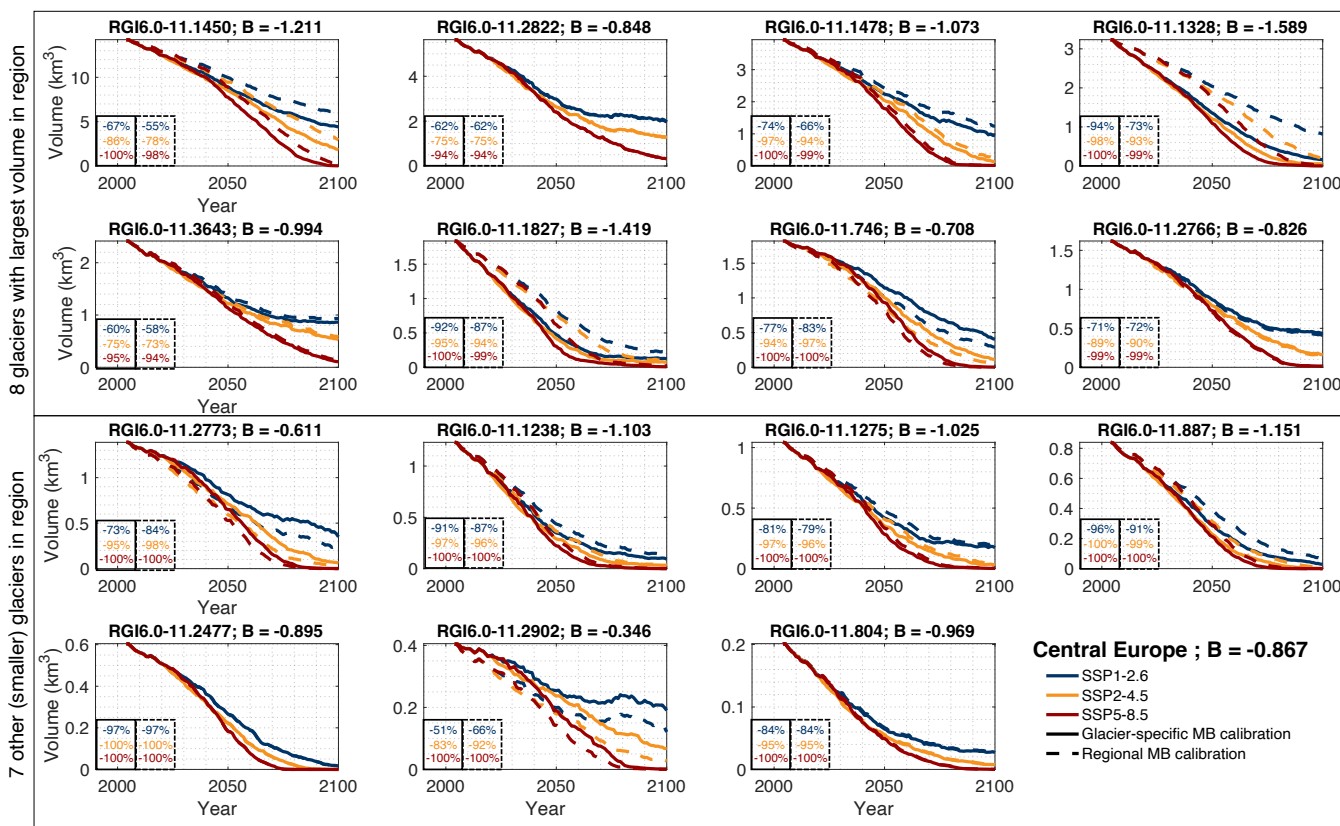


**Figure 2:** Projected glacier evolution with GloGEM for 15 glaciers in RGI region 11 (Central Europe) when calibrated to
glacier-specific mass balances (full line) and the regional mass balance (dotted line) under various future climate projections
(multi climate-model median shown here for every SSP). The two upper rows of panels represent the eight largest glaciers in
the region, while the lowest two rows are sampled from seven other glaciers in the region to cover the glacier volume range.

The title of every panel is the RGIv6.0 glacier ID and the corresponding glacier-specific mass balance (B) for the period 2000-





2019 in m w.e. a⁻¹ (from Hugonnet et al., 2021). Note that the y-axis scale differs among the panels. For every glacier (panel), the numbers in the lower left corner correspond to the 2015-2100 volume change, when calibrated to the glacier-specific mass balances (full box; left) and the regional mass balance (dotted box; right). For visual clarity, the results are here shown for selected SSPs (SSP1-2.6, SSP2-4.5, and SSP5-8.5). The calibrated mass balance parameters for these 15 glaciers are shown in
Figure 3.

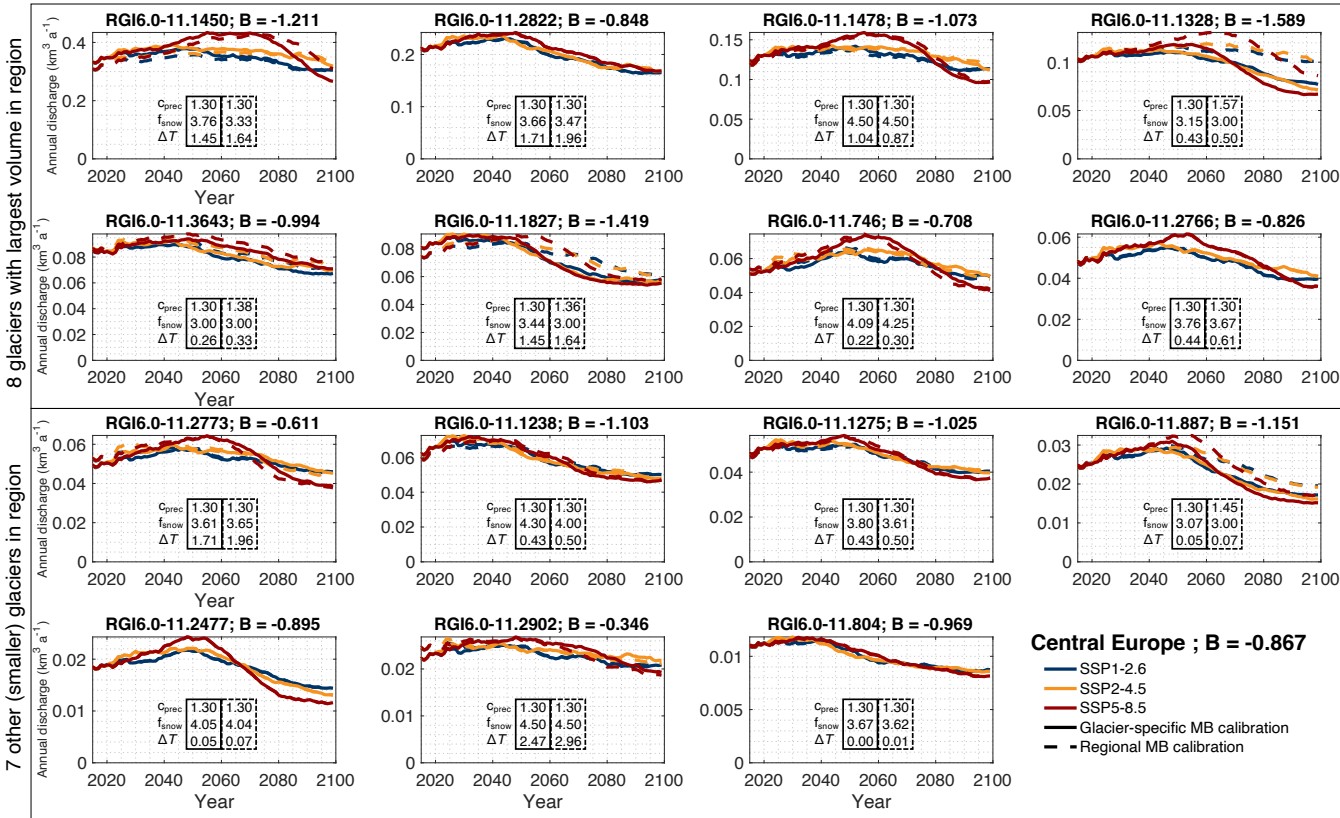

**Figure 3**: Projected annual glacier discharge in km³ a⁻¹ as modelled with GloGEM for 15 glaciers in RGI region 11 (Central Europe) when calibrated to glacier-specific mass balances (full line) and the regional mass balance (dotted line) under various future climate projections (multi climate-model median shown here for every SSP; 20-yr running average filter). The two
upper rows represent the eight largest glaciers in the region, while the lowest two rows are sampled from seven other glaciers in the region to cover the glacier volume range. The title of every panel is the RGIv6.0 glacier ID and the corresponding glacier-specific mass balance (B) for the period 2000-2019 in m w.e. a⁻¹ (from Hugonnet et al., 2021). Note that the y-axis scale differs among the panels. For visual clarity, the results are here shown for selected SSPs (SSP1-2.6, SSP2-4.5, and SSP5-8.5). For every glacier (panel), the calibrated values for $c_{prec}$, $f_{snow}$, and $\Delta T$, are shown for the glacier-specific calibration (full
box; left) and the regional mass calibration (dotted box; right). The range over which $c_{prec}$ can vary differs regionally (for the European Alps this is between 1.3 and 2.3, see Table S 2).



### 4.1.2 Regional scale

Generally, the differences in projected volume change at the glacier-specific scale (Figure 2) largely even out at the regional
scale: the glaciers that have a more positive mass balance (vs. regional average) are projected to lose less mass, which is
compensated by glaciers with a more negative mass balance, translating into a stronger future loss. As a consequence,
differences in the 2015-2100 regional volume change projections are generally small (Figure 4), i.e., within 3% for most
regions and climate scenarios. This is for instance the case for Western Canada and the United States (RGI region 2; Figure
5a), where the differences in 2015-2100 regional volume change arising from the calibration approach are <2% under all
climate scenarios (Figure 4). In this region, the volume change is slightly more pronounced with the glacier-specific calibration.
This very subtle difference is due to the fact that for Western Canada and the United States, under extreme warming, the little
remaining regional ice volume is typically located in the largest glaciers (which take longer to disappear), which here tend to
have a mass balance that is more negative than the regional one (Figure 5a; see signal from largest glaciers/symbols).

In various polar regions, there is a slight tendency for the glacier-specific calibration to result in less loss than the regional
calibration. This difference is the most apparent in RGI region 7 (Svalbard) and RGI region 19 (Antarctic and subantarctic),
where the regional calibration results in 2015-2100 volume losses that are 4 to 6% greater compared to the glacier-specific
calibration (Figure 4). This difference partly originates from the glaciers that are most resistant to warming, i.e. the glaciers
that lose least mass when forced to a similar mass balance (e.g., the regional mass balance), which tend to have a mass balance
that is less negative (more positive) than the regional one (Figure 5b; glaciers in the right upper side). As a consequence, the
absolute volume difference that arises from the two calibration approaches for these glaciers outweighs the signal from the
glaciers with a more negative mass balance (Figure 5b; glaciers in the lower left side). A similar mechanism is at play in other
polar regions (Arctic Canada North, Greenland Periphery, Svalbard, Russian Arctic), where the glaciers that are the most
resistant to warming (i.e. least relative loss when calibrated to regional mass balance) typically also have a glacier-specific
mass balance that is less negative (more positive) than the regional one (Figure S 1), resulting in a more pronounced loss under
the regional calibration (Figure 4). Additionally, the less negative (more positive) mass balance of these more resistant glaciers
allows these ice bodies to lose less mass (partly grow) and suffer less (profit more) from the mass balance – elevation feedback.





**Figure 4:** Evolution of 21st century regional glacier volume as modelled with GloGEM when the mass balance component is calibrated to match glacier-specific observations (full line) and regional observations (dotted line) (all mass balance observations from the Hugonnet et al., 2021 dataset). Multi climate-model median shown for every SSP (for a quantification of the spread around these values and more in-depth focus on the glacier evolution as opposed to differences related to model calibration, refer to Figure 7, Figure 8 and section 4.2). For visual clarity, the results are here shown for selected SSPs (SSP1-2.6, SSP2-4.5, and SSP5-8.5).





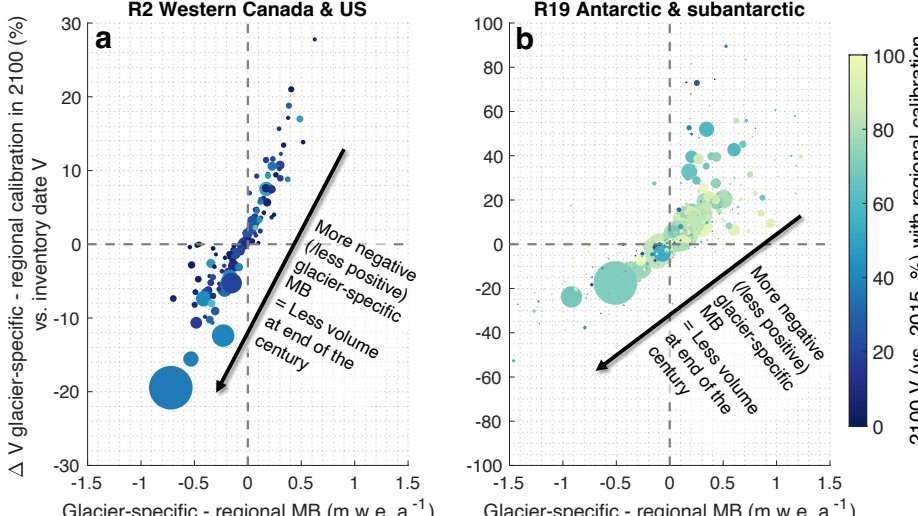

**Figure 5**: Difference in the projected volume change (between inventory date and 2100 under SSP2-4.5) for the glacier-specific vs. regional mass balance calibration under SSP2-4.5 (multi-climate model median shown here). Here, two RGI regions are highlighted: (a) Western Canada and US (RGI region 2) and (b) Antarctic and subantarctic (RGI region 19), while other regions are shown in the suppl. mat. (Figure S 1). Every dot represents an individual glacier with a volume >1km³, where the size of the dot relates to the glacier consensus volume estimate (Farinotti et al., 2019a) and the colour represents the 2100 volume change (vs. 2015) for the projections with the regional calibration (i.e., same mass balance forcing for every glacier). Note that the y-axis scale differs among the two panels.

### 4.1.3 Global scale

At the global level, we find that the projected evolution is only slightly affected by the calibration data, with mass loss differences over the period 2015-2100 being around 3% under all SSPs (Figure 4). This global difference, with slightly less loss when the model is calibrated to glacier-specific mass balance observations arises primarily from the signal from the Antarctic and subantarctic glaciers (RGI region 19), since this is the most voluminous region on Earth and the one for which differences resulting from the calibration technique are the most prevalent (Figure 4). From these findings, we argue that the largest added value of relying on the glacier-specific calibrations is not for regional and global projections but rather at the glacier level. However, we do note that for the regional to global level, the effect of the calibration strategy on projections is not negligible, and that when considering the long-term evolution of glaciers (i.e. post 2100), differences between both calibration approaches could become larger, particularly for large glaciers that considerably differ from the regional mean. At the glacier level, the mass balance calibration with glacier-specific data increasingly allows projecting the glacier-specific evolution around the globe and assessing related impacts (e.g. related to runoff, as highlighted in Figure 3).





## 4.2 Regional and global 21$^{st}$ century glacier evolution under CMIP6 scenarios

Here, we extend the GloGEM simulations that were calibrated to match the glacier-specific mass balance with new simulations performed with the OGGM v1.6.1 (Schuster et al., 2023b). Through these GloGEM and OGGM simulations, we provide new

estimates on the global glacier evolution under CMIP6 and thereby expand the recent estimates obtained from the coupled PyGEM model by Rounce et al. (2023). In order to allow for a direct comparison, the GloGEM and OGGM simulations are run with the same future climatic forcing as the PyGEM simulations by Rounce et al. (2023).

### 4.2.1 GloGEM and OGGM

At present (2020-2025), annual global glacier losses modelled with GloGEM and OGGM are around 350-400 km$^3$ yr$^{-1}$.

Assuming a density conversion of 900 kg m$^{-3}$ for ice (Cuffey and Paterson, 2010), these losses logically agree well with Hugonnet et al. (2021)'s observations (e.g., 298±24 Gt yr$^{-1}$ for 2015-2019 time period) to which our models are calibrated. The annual global glacier loss is projected to increase in the coming 10-15 years, irrespective of the climate scenario, reaching losses for the year 2035 of around 450-550 km$^3$ yr$^{-1}$ for GloGEM to 550-600 km$^3$ yr$^{-1}$ for OGGM (Figure 6). After this, the losses become more scenario-dependent, but by the mid-century the differences in global glacier evolution arising from

different climatic scenarios are still very limited: GloGEM projects 2015-2050 volume loss between 11±7% (SSP1-2.6) and 13±7% (SSP5-8.5), while OGGM projects losses between 12±5% (SSP1-2.6) and 14±6% (SSP5-8.5) (Figure 7; multi-climate model median, with 95% confidence interval).

Under a low-emission scenario (SSP1-2.6), annual losses decrease throughout the second half of the century, and eventually

reach values of 300-400 km$^3$ yr$^{-1}$ by 2100 (Figure 6), which is close to current losses. In contrast, under high-emission scenarios, annual losses unabatedly increase and by the late 21$^{st}$ century reach about 1100-1200 km$^3$ yr$^{-1}$ (GloGEM; Figure 6a) to 1300-1400 km$^3$ yr$^{-1}$ (OGGM; Figure 6b) under SSP5-8.5, i.e. values that are about three times as large as current losses. Consequently, the impact of the climate scenario is very pronounced at the end of century: under SSP1-2.6 the 2015-2100 global glacier volume is projected to decrease by 25±15% (GloGEM; multi-climate model median; Figure 7) to 29±14%

(OGGM), while under SSP5-8.5 losses of 46±26% (GloGEM) to 54±29% (OGGM) are projected.



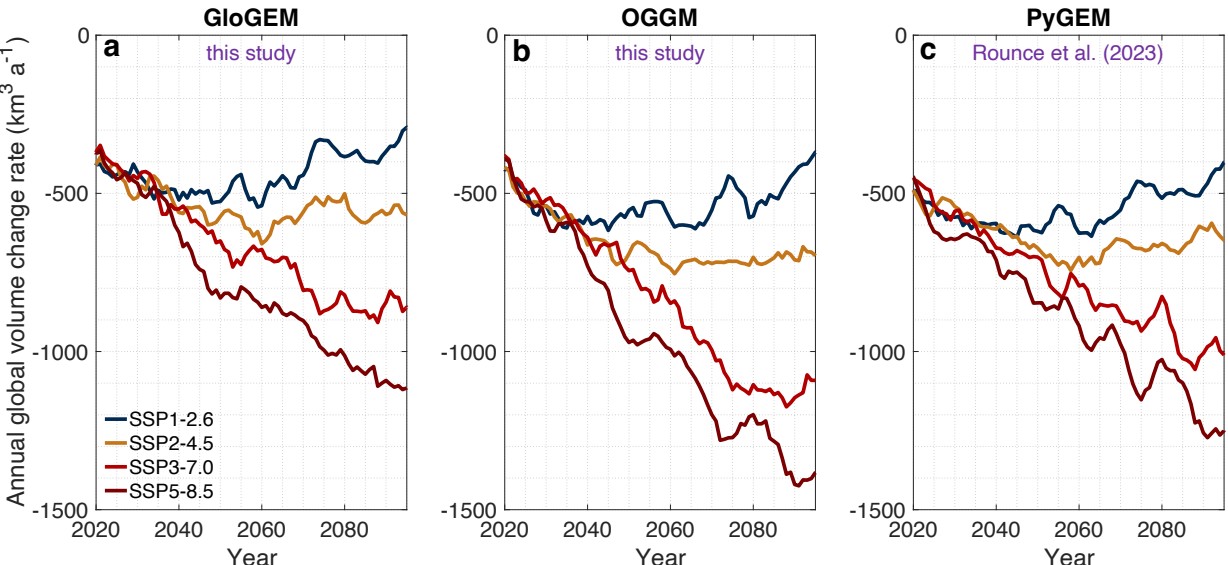

**Figure 6**: Annual global glacier volume change rate as projected with (a) GloGEM (this study), (b) OGGM (this study), and (c) PyGEM (Rounce et al., 2023) under various future climate scenarios (multi climate-model median shown for every SSP, filtered with 5-yr running mean).

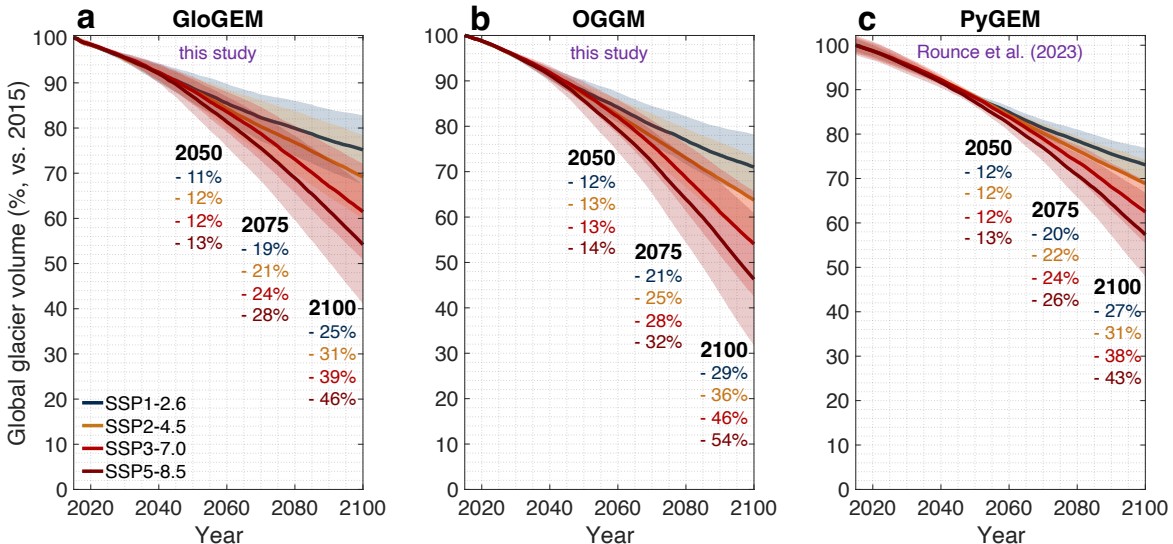

**Figure 7:** Evolution of 21st century global glacier volume compared to 2015 as modelled with (a) GloGEM (this study), (b) OGGM (this study), and (c) PyGEM (Rounce et al., 2023) under various future climate scenarios (multi climate-model median shown for every SSP). Shading indicates ±1 standard deviation of climate model ensemble. As opposed to GloGEM and OGGM, for PyGEM (Rounce et al., 2023) the initial volume is dependent on the climate scenario, hence the spread in projected global glacier volume from 2015 onwards.



At the global level, OGGM projects more global volume loss than GloGEM, with a 4% (SSP1-2.6) to 8% (SSP5-8.5) higher
2015-2100 volume loss (vs. GloGEM, Figure 7; Table 1). This difference mainly results from the projections for the Antarctic
and subantarctic glaciers (RGI region 19), which is the largest of the RGI regions (RGI Consortium, 2017; Farinotti et al.,
2019a), and where differences in projected ice volume significantly differ under all climate scenarios (t-test, 1% significance
level; Table 1). For the Antarctic and subantarctic glaciers, GloGEM projects a 2015-2100 mass loss of 14±13% to 33±24%,
while with OGGM this loss varies between 21±18% to 52±32% (range is multi-climate model median between SSP1-2.6 and
SSP5-8.5, respectively; Figure 8; Table 1). A part of this projected difference is linked to the inclusion of frontal ablation,
which is represented through a simplified approach in GloGEM (based on Oerlemans and Nick, 2005), whereas frontal ablation
is not explicitly represented in the OGGM setup that we utilize here (new representations of frontal ablation exist for OGGM,
e.g. Malles et al., 2023, but are not available for Antarctica). In GloGEM, frontal ablation contributes to the total mass balance
and thereby increases the surface mass balance (vs. case without frontal ablation). As a consequence, if future frontal ablation
decreases (e.g. loss contact with ocean), this more positive mass balance dominates and causes less future ice loss (vs. case in
which frontal ablation is not represented and the glaciers are more sensitive to changes in temperature). Given the very large
uncertainties in modelled present-day and future frontal ablation, it is currently difficult to judge whether results from a setup
with a relatively uncertain frontal ablation (GloGEM) or one in which it is not explicitly represented (OGGM setup used in
this study) should be more trusted.

In other polar regions, the projected mass loss is generally relatively similar for both models, with a minor (non-significant)
tendency for most regions to have a stronger mass loss in OGGM (Arctic Canada North, Arctic Canada South, Greenland
Periphery, Iceland, Svalbard), except for Russian Arctic, where GloGEM projects slightly more loss (Figure 8, Figure 10,
Table 1). The relative similarity in projected changes for polar regions is also apparent from the similar global volume evolution
when excluding the Antarctic and subantarctic glaciers (i.e., summing glacier changes over RGI regions 1 to 18; Figure S 2 in
suppl. mat.): in this case GloGEM projects a 2015-2100 mass loss of 30±18% to 52±28%, while OGGM projects losses of
32±20% to 54±41% (multi-climate model median between SSP1-2.6 and SSP5-8.5, respectively).

For mountain glaciers, various regions have similar projections, with insignificant differences for all climate scenarios for
Alaska, North Asia, Central Europe, Central Asia, South Asia West, and New Zealand. In other mountain regions, some
significant differences exist, with OGGM projecting a larger loss for Western Canada & US (under SPP1-2.6 and SSP2-4.5)
and Scandinavia (all SSPs), whereas GloGEM projects stronger losses for Caucasus (SSP1-2.6), South Asia East (SPP1-2.6
and SSP2-4.5), Low Latitudes (SPP1-2.6 and SSP2-4.5), and Southern Andes (SSP3-7.0 and SSP5-8.5) (Table 1).




**Figure 8:** Evolution of 21st century glacier volume compared to 2015 as modelled with GloGEM and OGGM for every region of the Randolph Glacier Inventory (RGI v6.0) under various future climate projections (multi climate-model median shown for every SSP). Shading indicates ±1 standard deviation of climate model ensemble. In these simulations, the mass balance forcing component is calibrated for every glacier to match the glacier-specific geodetic mass balance observations by Hugonnet et al. (2021). Similar plots are presented for the PyGEM simulations by Rounce et al. (2023) in Figure S 3, whereas the 2015-2100 projected changes are directly compared for the three glacier models in Table 1 and Figure 10.





| RGI | Region | GloGEM (this study) | | | | OGGM v1.6.1 (this study) | | | | PyGEM (Rounce et al., 2023) | | | |
|---|---|---|---|---|---|---|---|---|---|---|---|---|---|
| | | SSP1 2.6 | SSP2 4.5 | SSP3 7.0 | SSP5 8.5 | SSP1 2.6 | SSP2 4.5 | SSP3 7.0 | SSP5 8.5 | SSP1 2.6 | SSP2 4.5 | SSP3 7.0 | SSP5 8.5 |
| 1 | Alaska | 49±21 | 61±25 | 66±23 | 75±27 | 49±20 | 59±26 | 67±25 | 73±29 | 51±16 | 58±20 | 66±20 | 72±24 |
| 2 | Western Canada & US | 74±19 | 86±15 | 94±11 | 97±10 | 84±14 | 94±10 | 98±4 | 99±4 | 87±13 | 95±9 | 99±5 | 99±3 |
| 3 | Arctic Canada North | 11±16 | 14±20 | 17±28 | 21±37 | 16±17 | 20±22 | 26±31 | 29±40 | 16±7 | 18±11 | 21±17 | 24±26 |
| 4 | Arctic Canada South | 35±19 | 42±21 | 49±23 | 54±28 | 42±15 | 52±18 | 60±22 | 66±28 | 45±15 | 52±17 | 58±21 | 65±27 |
| 5 | Greenland periphery | 26±19 | 32±21 | 41±25 | 47±31 | 34±21 | 41±22 | 48±27 | 55±33 | 33±13 | 38±15 | 44±21 | 49±27 |
| 6 | Iceland | 25±34 | 33±37 | 44±35 | 49±35 | 35±24 | 44±26 | 50±25 | 58±26 | 39±22 | 47±25 | 54±25 | 61±27 |
| 7 | Svalbard | 26±34 | 36±35 | 45±34 | 59±33 | 29±31 | 40±38 | 56±37 | 73±33 | 29±21 | 37±27 | 48±32 | 63±30 |
| 8 | Scandinavia | 43±26 | 55±27 | 66±24 | 71±24 | 68±16 | 79±17 | 88±11 | 90±10 | 75±18 | 87±18 | 94±10 | 96±7 |
| 9 | Russian Arctic | 23±24 | 27±29 | 37±30 | 48±30 | 19±19 | 21±28 | 29±34 | 37±37 | 22±12 | 26±19 | 35±23 | 42±26 |
| 10 | North Asia | 74±14 | 84±11 | 90±8 | 92±6 | 78±10 | 88±7 | 92±5 | 94±5 | 81±10 | 89±6 | 93±5 | 95±4 |
| 11 | Central Europe | 78±12 | 91±7 | 97±4 | 99±3 | 81±11 | 93±6 | 98±3 | 99±1 | 86±8 | 95±4 | 99±2 | 99±1 |
| 12 | Caucasus | 76±10 | 88±6 | 94±3 | 97±3 | 67±11 | 85±6 | 95±3 | 98±3 | 72±9 | 87±5 | 95±3 | 98±2 |
| 13 | Central Asia | 55±13 | 67±11 | 79±10 | 88±10 | 53±13 | 68±9 | 79±9 | 88±10 | 50±10 | 62±7 | 72±7 | 80±9 |
| 14 | South Asia West | 38±29 | 49±29 | 61±27 | 74±27 | 43±20 | 53±21 | 64±20 | 76±22 | 40±14 | 48±16 | 57±17 | 67±19 |
| 15 | South Asia East | 75±17 | 86±12 | 92±7 | 95±6 | 63±17 | 79±12 | 88±8 | 93±8 | 77±12 | 85±8 | 91±5 | 94±5 |
| 16 | Low Latitudes | 80±19 | 94±9 | 98±4 | 99±3 | 67±24 | 88±11 | 96±5 | 97±4 | 77±18 | 91±9 | 97±4 | 99±3 |
| 17 | Southern Andes | 47±28 | 56±28 | 67±27 | 74±27 | 41±15 | 46±15 | 54±19 | 59±20 | 49±19 | 57±19 | 68±21 | 73±22 |
| 18 | New Zealand | 52±23 | 70±17 | 84±14 | 87±12 | 41±27 | 66±17 | 83±14 | 86±11 | 64±22 | 81±12 | 93±8 | 94±7 |
| 19 | Antarctic and subantarctic | 14±13 | 17±17 | 26±18 | 33±24 | 21±18 | 28±22 | 42±24 | 52±32 | 14±4 | 16±5 | 21±7 | 25±11 |
| | **Global** | 25±15 | 31±18 | 39±21 | 46±26 | 29±14 | 36±19 | 46±23 | 54±29 | 27±8 | 31±11 | 38±14 | 43±19 |

**Table 1:** Relative regional and global glacier volume loss over the 2015-2100 period (in %) as modelled in this study with GloGEM and OGGM, and in Rounce et al. (2023; PyGEM). Values are the multi-climate model median and the corresponding 95% confidence intervals (see Text S1 for calculation). Colour coding indicates cases where the differences are statistically significant (t-test, 1% significance level) with respect to another glacier model, where the colour refers to the glacier model against which the difference is significant. When the projection is significantly different compared to the two other glacier models, this is highlighted in dark orange.

**4.2.2 Towards a CMIP6 global glacier evolution ensemble: comparison with PyGEM (Rounce et al., 2023)**

Our modelled regional and global glacier volume changes agree well with those from Rounce et al. (2023; hereafter referred to as "PyGEM"). At the global scale, the 2015-2100 projected PyGEM losses are close to those we project here with GloGEM (Figure 9), with SSP1-2.6 losses of 25±15% (GloGEM) and 27±8% (PyGEM), and SSP5-8.5 losses of 46±26% (GloGEM) and 43±19% (PyGEM). The difference in projected mass loss is more scenario-dependent in GloGEM, with a difference in the 2015-2100 mass loss between SSP1-2.6 and SSP5-8.5 of 21% (GloGEM) vs. 16% (PyGEM). For OGGM, the SSP-dependence is the most pronounced, with a 25% difference in 2015-2100 mass loss between SSP1-2.6 and SSP5-8.5, while the total mass loss is slightly more pronounced than for PyGEM. Table 1Figure 7A noteworthy distinction is that in PyGEM the uncertainty is greatly reduced around 2040, which results from the initialization of PyGEM that accounts for differences in climate models (resulting in a different initial volume), while for OGGM and GloGEM, at initialisation, all projections start





at the same volume. In PyGEM simulations, the climate model that has the largest initial volume results in the smallest volume at the end (and vice-versa), so the simulations converge around 2040 resulting in relatively small absolute uncertainties (Rounce et al., 2023).

The global differences in the projected global losses for the three models are mostly determined by the regional evolution of the Antarctic and subantarctic glaciers (RGI region 19), where the projected losses with PyGEM and GloGEM are relatively similar, and thus less pronounced than for OGGM (Table 1 and Figure 10; see also GloGEM vs. OGGM comparison in section 4.2.1). A direct comparison is difficult, since processes such as frontal ablation are accounted for differently, and also the initial absolute volume difference differs, since Rounce et al. (2023) set up PyGEM to match the land-terminating ice volume

from Farinotti et al. (2019a), after which the ice thickness inversion that included frontal ablation was performed which increased the initial ice volume. Other differences relate to the mass balance model, where GloGEM and PyGEM have a similar architecture with a surface-type distinction and rely on variable temperature lapse rates (from ERA5), while the utilized OGGM setup does not distinguish surface types and uses default temperature lapse rates, which impacts projected losses (Schuster et al., 2023a). Another difference relates to the evolution framework, which accounts for ice dynamics in OGGM

and PyGEM, whereas GloGEM does not explicitly represent ice dynamical processes and evolves its glacier geometry through a retreat parameterization.

   Despite these differences in model setup and architecture, in the majority of regions, the projected changes with GloGEM and OGGM are close to the PyGEM projections (Figure 10; Table 1). In 14 RGI regions, the PyGEM projections are similar to

those from GloGEM and OGGM presented here, with insignificant differences (under all climate scenarios) to both models (Alaska, Arctic Canada North, Greenland periphery, Iceland, Svalbard, Russian Arctic), GloGEM (South Asia West, South Asia East, Low Latitudes, Southern Andes, Antarctic and subantarctic), or OGGM (Western Canada & US, Arctic Canada South, North Asia). In the other five RGI regions, PyGEM is significantly different to the two other models, with cases where the mass loss is consistently higher (Scandinavia (SSP3-7.0 and SSP5-8.5), Central Europe (SSP1-2.6), New Zealand (all

SSPs)), consistently lower (Central Asia (SSP2-4.5 to SSP5-8.5)), or consistently in between both models (Caucasus (SSP1-2.6)).




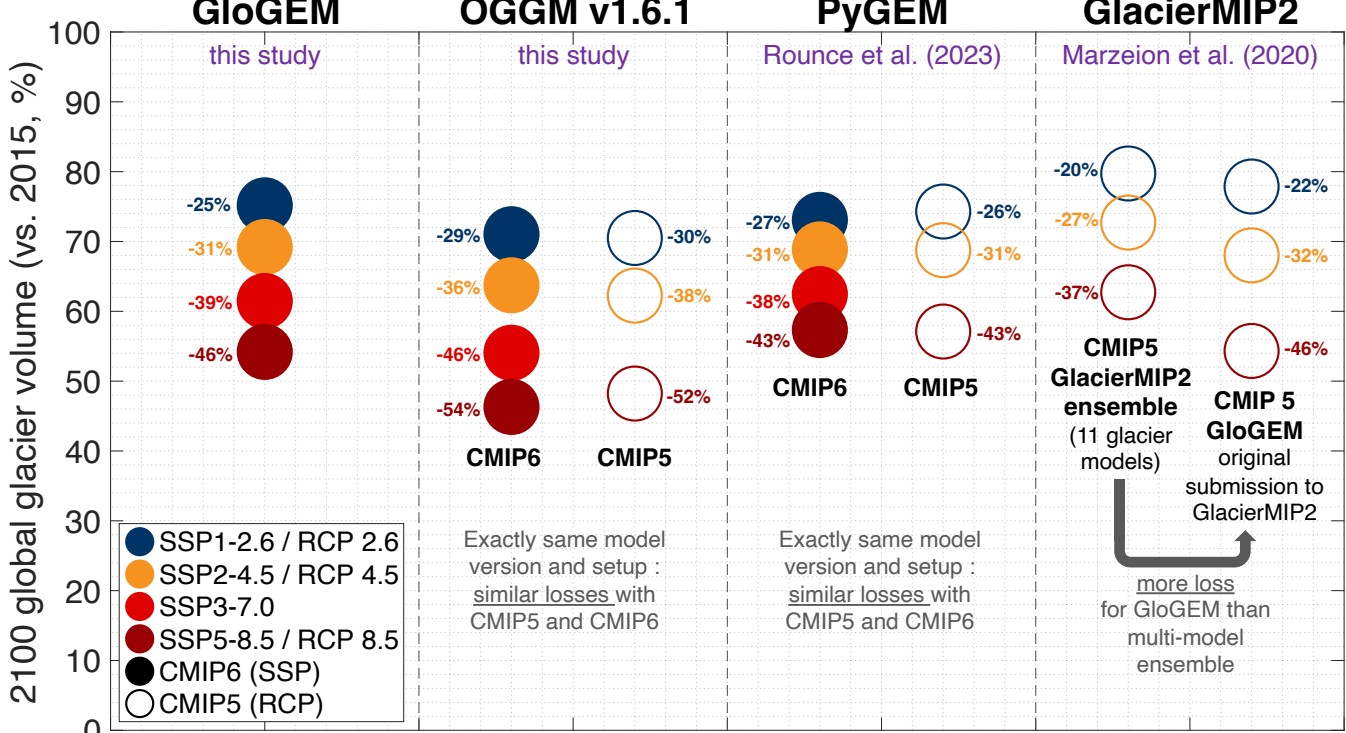

**Figure 9:** Comparison of global 2015-2100 volume change (multi-climate model median) as modelled in this study (GloGEM

and OGGM; Figure 7), with PyGEM (Rounce et al., 2023), and in GlacierMIP2 (Marzeion et al., 2020). The solid circles
represent the evolution as modelled with CMIP6 climate models under SSP scenarios, whereas the transparent symbols
correspond to the evolution as modelled with the CMIP5 climate model ensemble that was used in GlacierMIP2 (Marzeion et
al., 2020) under RCP scenarios. For GloGEM, the original CMIP5 GlacierMIP2 simulation is shown (i.e., part of the
GlacierMIP2 ensemble), whereas for OGGM and PyGEM the CMIP5 simulations were re-run with the same setup as the

CMIP6 simulations.

**4.2.3 Comparison to the second phase of the Glacier Model Intercomparison Project (GlacierMIP2)**

We also compare our projected glacier evolution to the current community estimate of global glacier change which was
performed within the second phase of the Glacier Model Intercomparison Project (GlacierMIP2; Marzeion et al., 2020), in
which eleven glacier models of varying complexity were used to model the regional to global scale evolution of glaciers. This

comparison is not straightforward, since GlacierMIP2 relied on the climate model simulations from the CMIP5 ensemble
(Taylor et al., 2012), which consist of different $CO_2$-emission scenarios and different climate models compared to the CMIP6
ensemble that we consider here. For a given radiative forcing level (e.g., 2.6 W/m$^2$, corresponding to RCP2.6 and SSP1-2.6),
under all scenarios, the projected 2015-2100 CMIP6 loss is substantially higher with GloGEM, OGGM, and PyGEM compared
to the GlacierMIP2 CMIP5 ensemble (Figure 9): with 2015-2100 volume differences ranging from 5% (SSP1-2.6/RCP2.6: -



25% loss in GloGEM vs. -20% in GlacierMIP2) to 17% (SSP5-8.5/RCP8.5: -54% loss in OGGM vs. -37% in GlacierMIP2).
The differences are particularly pronounced in Alaska, Western Canada & US, Arctic Canada South, High-Mountain Asia
(Central Asia, South Asia West, South Asia East), and Southern Andes, for which, for a given radiative level, the CMIP6
losses for all three individual models (GloGEM, OGGM, and PyGEM) are larger than the CMIP5 GlacierMIP2 losses (Figure
10).


Whereas a direct and in-depth comparison is not straightforward, relating GloGEM, OGGM, and PyGEM simulations forced
with the same CMIP5 ensemble as GlacierMIP2 offers insights in a part of the projected differences. Generally, temperatures
in CMIP6 climate simulations are known to be more sensitive to radiative forcing than in CMIP5, resulting in higher
temperatures for same radiative forcing levels (Tokarska et al., 2020; Hausfather et al., 2022). However, for the ensembles
considered here (CMIP5 GlacierMIP2 ensemble vs. CMIP6 ensemble used in our study and in Rounce et al. (2023)),
temperatures are relatively similar, resulting in very similar losses for CMIP6 compared to CMIP5 for the OGGM and PyGEM
simulations (Figure 9). To understand the effect of the glacier model on the differences, the original GlacierMIP2 study offers
interesting insights, where GloGEM and OGGM projected losses that were substantially larger than the multi-model median
(shown in Figure 9 for GloGEM; not shown for OGGM, which was not run globally in GlacierMIP2). Newly performed
OGGM and PyGEM CMIP5 simulations (with exactly same model setup as CMIP6 simulations; not available from GloGEM)
support this higher model-specific loss (Figure 9), although differences in actual model setup hinder a direct comparison (e.g.,
different mass balance data used for calibration in GlacierMIP2 (e.g. Gardner et al., 2013; Zemp et al., 2019) compared to new
simulations relying on Hugonnet et al. (2021)). For OGGM, the new CMIP5 projections (v1.6.1) and the older ones submitted
to GlacierMIP2 are very similar at the regional scale (except for Svalbard, where the loss is more limited in the newest version;
and some regions where the differences between the scenarios was less pronounced in GlacierMIP2: High-Mountain Asia,
Low Latitudes, Southern Andes, New Zealand), indicating a limited effect of changes in model architecture, input, and
calibration data on projected changes (Figure 10). From these comparisons, we conclude that an important part of our higher
projected loss results from the selected set of glacier models, rather than from differences in climate forcing and/or other
changes in model parameters and input (in our study vs. in GlacierMIP2). Therefore, we expect models that predicted less
glacier loss in GlacierMIP2, i.e. mostly the more simplified volume-area scaling models (some of which rely on energy-balance
modelling), to likely predict a reduced mass loss compared to our new projections (GloGEM and OGGM) and those by Rounce
et al. (2023; PyGEM) when forced with the same CMIP6 ensemble.





**Figure 10:** Comparison of global 2015-2100 volume change (multi-climate model median) as modelled in this study (GloGEM and OGGM; Figure 8), with PyGEM (Rounce et al., 2023), and in GlacierMIP2 (Marzeion et al., 2020). The solid circles represent the evolution as modelled with CMIP6 climate models under SSP scenarios, whereas the transparent symbols correspond to the evolution as modelled with the CMIP5 climate model ensemble that was used in GlacierMIP2 (Marzeion et al., 2020) under RCP scenarios. We here also show the GloGEM and PyGEM submission to GlacierMIP2, which were not included in Figure 9 since the models did not have a global coverage (available for 18 and 3 RGI regions, respectively).



## 5. Conclusions

The calibration of glacier evolution models is of large importance since it directly determines the modelled glacier sensitivity to changing climatic conditions. Whereas up to a few years ago global glacier models could only be constrained based on scattered mass balance observations on a few glaciers or regional mass balance estimates, new datasets now allow for a calibration of models to match glacier-specific observed changes for every glacier on Earth. Our study quantifies how calibrating glacier evolution models to match glacier-specific geodetic mass balances influences the projected glacier evolution. This comparison is important since glacier-specific mass balance observations such as those provided by Hugonnet et al. (2021) are now becoming the new standard when calibrating the mass balance component in global glacier evolution models. Our analysis isolates the effect that glacier-specific mass balance observations have on projections at various spatial scales, highlighting that:

- At the glacier-specific scale, the type of mass balance observation (glacier-specific vs. regional) used for model calibration has a substantial effect on modelled future changes. For some glaciers, differences in the projected 2015-2100 volume loss can be on the order of tens of percent. These differences at the glacier level are very apparent in some regions, e.g., in High-Mountain Asia, 35-55% of all glaciers have differences in the modelled 2015-2050 volume changes that exceed 10% depending on the data used for model calibration. These pronounced differences suggest that a calibration to glacier-specific mass balance observations now also progressively allows for global-scale model results to be used for small-scale applications and for assessing local glacier impacts. In combination with future glacier model advances and an integration of additional types of observations, this will increasingly allow quantifying the water supply from glaciers in small catchments, or glacier-related hazards, both of which strongly depend on the glacier-specific evolution.

- At regional to global scales, the effect of the calibration strategy on future projections is generally more limited, but not negligible. Projected 2015-2100 volume differences between both approaches are around 3% globally, and up to 6-7% at a regional scale in the most extreme cases. These differences in projected glacier changes mostly arise from the signal from the glaciers that are most resistant to warming in the 21$^{st}$ century. When considering the longer-term glacier evolution (i.e. multi-century), the effect of the calibration strategy on future projections could become important, particularly for large glaciers for which the mass balance considerably differs from the regional mean.

Additionally, our newly performed simulations contribute to creating an ensemble of global glacier projections under CMIP6 scenarios by complementing the first global CMIP6 glacier projections by Rounce et al. (2023; with PyGEM)

In our GloGEM and OGGM simulations, we project that the annual global glacier volume loss is to increase under all climatic scenarios until 2035, reaching values that are about 30-70% more than present-day (2020-2025) losses. Throughout the second half of the century, under a low-emission scenario (SSP1-2.6), annual losses decrease and eventually reach present-day levels



by 2100, resulting in a global volume loss of around 25-29% (2100 vs. 2015). In contrast, under high-emission scenarios, we
project annual losses to continuously increase until 2100, peaking around three times current losses, and translating into losses
of 46-54% over the 2015-2100 time period. These projected 21$^{st}$ century losses generally agree well with those simulated with
PyGEM by Rounce et al. (2023). Despite some regional differences and a slightly higher projected sensitivity of glacier loss
to climatic conditions in our simulations, a good agreement exists globally, thereby confirming Rounce et al. (2023)'s larger
mass loss compared to the GlacierMIP2 values for a given radiative forcing level (Marzeion et al., 2020). Our analysis suggests
that the larger loss mainly originates from the considered models (GloGEM, OGGM, PyGEM), rather than the differences in
model input (geometry and mass balance) and climatic forcing (CMIP5 vs. CMIP6 ensemble).

In the coming years, a more comprehensive picture of global glacier evolution under the latest CMIP6 scenarios will be
obtained as other glacier models will be forced with the same climatic forcing, thereby further expanding the current ensemble.
In this respect, new glacier outlines (RGI v7.0, RGI Consortium, 2023) and other datasets with a (near) global coverage (e.g.,
on ice surface velocities and ice thickness reconstructions (e.g. Millan et al., 2022), and frontal ablation estimates (Kochtitzky
et al., 2022)) will allow inverting glacier properties, calibrating model parameters, and evaluating model performance in a
more advanced way, increasingly relying on machine learning and data assimilation (e.g. Bolibar et al., 2023; Cook et al.,
2023). These advances are likely to improve the credibility of future glacier projections at various spatial scales (from glacier-
specific to global) and spanning over a range of time scales (from decadal to multi-centennial).

**Data availability**

The GloGEM simulations as presented in this manuscript are available from Zekollari et al. (2024), while OGGM simulations
presented in this manuscript are available from Schuster et al. (2023b). The PyGEM-OGGM simulations are available at
https://nsidc.org/data/hma2_ggp/versions/1.

**Author contributions**

HZ designed the study, performed the analyses of the simulations, wrote the manuscript, and made the figures, in direct
collaboration with MH and DF. MH ran the GloGEM simulations, while LS and FM ran the OGGM simulations. LS and RA
assisted with data processing. All co-authors were involved in the scientific discussions that lead to this study, with major and
detailed contributions from MH, LS, FM, RH, BM, DR, and DF. All authors provided feedback on the manuscript.

**Competing interests**

Ben Marzeion and Daniel Farinotti are members of the editorial board of The Cryosphere.



**Financial support**

HZ, MH, DF, BM, NC, and SM were supported by the European Union's Horizon 2020 research and innovation programme (PROTECT; grant agreement number 869304). HZ and RA acknowledge the funding received from the European Research Council (ERC) under the European Union's Horizon Framework research and innovation programme (grant agreement No 101115565; 'ICE$^3$' project). HZ was also supported through a postdoctoral fellowship (chargé de recherches) of the Fonds de la Recherche Scientifique (FNRS) and by the research foundation – Flanders (FWO) through an Odysseus Type II project (grant agreement G0DCA23N; 'GlaciersMD' project). LS is recipient of a DOC Fellowship of the Austrian Academy of Sciences at the Department of Atmospheric and Cryospheric Sciences, Universität Innsbruck (No. 25928). LS and FM have received funding from the European Union's Horizon 2020 research and innovation programme under grant agreement (PROVIDE; No. 101003687). This text reflects only the author's view and the Agency is not responsible for any use that may be made of the information it contains. DR was supported by National Aeronautics and Space Administration grant 80NSSC20K1296 and the Gulf Research Program of the National Academies of Sciences, Engineering, and Medicine under award number 2000013282. LC and RH benefited from financial support of the Swiss National Sciences Foundation (grant no. 184634, project "PROGGRES").

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
