# Peer review of "21st century global glacier evolution under CMIP6 scenarios and the role of glacier-specific observations"

_EGUsphere, 2024_

## Referee Comment (RC1)

**Review of *21st century global glacier evolution under CMIP6 scenarios and the role of glacier-specific observations* by Zekollari et al**

**General comments**

This is an interesting and well-designed paper by Zekollari et al, which provides a comparison of global glacier volume change under CMIP6. Additionally, the paper investigates the use of glacier-specific calibration techniques. Overall, the paper well written with the results thoroughly discussed. I have no major comments, but think the clarity of the text could be improved in certain places and have listed some minor comments below (including some adjustments to figures).

**Specific and technical comments**

L90 – I find the description of this methodology a bit confusing, please consider rephrasing

L136 – Please state the year which the ice thickness dataset represents

L168 – Please specify when the past climate starts (e.g. XXXX until 2020)

L171 – Please specify when the future climate ends

L180 – What is the debiasing procedure?

L192 – I think 'Alternatively' should be changed to 'Additionally'

L227 – The sentence beginning 'In general, if…' is phrased in a slightly confusing way and could be reworked for better clarity – maybe something like 'In general, the mass balance calibration parameters are calibrated to give a more negative SMB for glaciers which have a lower mass balance than the regional average. This translates…'

L284 – I would change the sentence 'For most other regions, these differences are even more outspoken..'. I do not think outspoken is a good choice of word, and this sentence is a bit dense and confusing to read. Could be changed to something like 'For most other regions, there is an even larger proportion of glaciers which show differences of more

than 10% in their 2015-2050 volume projections. For instance, in High mountain Asia…'

L288 – This could also be rephrased/ broken up into two sentences. Else, add '..independent of calibration methodology' or similar.

L299 – I do not understand what you mean by 'since discharge is calculated over initial glacier area'

Fig.2 – Please make the text in the 2015-2100 volume change boxes larger

Fig. 3 – Please also make the text large in this figure (for the calibrated values)

Figs 2 and 3: It could be beneficial to move some of the panels to the supplementary material, so that the remaining panels can be more easily seen – as of now, they are quite busy.

Fig.7 – If possible, please extend the width so that it takes up the full page width/ has the same dimensions as Fig.6

L424 – The explanation of why the inclusion of frontal ablation in GloGEM leads to less global volume loss does makes sense, but I had to read it a few times to understand. It would be good to add in a bit more detail here to make it extra clear.

L468 – Formatting error (sentence beginning with 'Table1Figure7A')

Fig. 9 – It would be nice to move the GloGEM panel nexto to the panel which compares CMIP5 GloGEM to CMIP model ensemble, for ease of comparing the GloGEM results with each other. Also include a 'CMIP6' label under the GloGEM results like in the other panels. It could additionally be beneficial to add results showing the mean of the CMIP6 forced GloGEM/OGGM/PyGEM simulations, to be compared to the CMIP5 ensemble mean. In this case, you could move the CMIP5 GloGEM results to the GloGEM panel and then have an 'ensemble mean/ CMIP5 vs CMIP6' panel.

L519 – Here you compare CMIP5 ensemble to your new CMIP6 results, which I think would be strengthened through the above changes to Fig. 9.

I understand the caveats to direct comparison that you explain in the text, but still think this would be a useful visual aid.

L600 – This is a very long sentence that could benefit from being broken up into a few parts

Fig. S3 – Make this figure take up the full page width; the panels are too small at the moment

---

## Referee Comment (RC2)

Review of Zekollari et al.: 21$^{\text{st}}$ century global glacier evolution under CMIP6 scenarios and the role of glacier-specific observations

This study investigates how mass balance data used for calibrating global glacier model projections affects model outcomes. Until recently, global glacier projections relied on either limited mass balance data from a small subset of glaciers or regional mass balance observations. Recent work has produced a data set of geodetic mass balances for all glaciers worldwide, thus enabling glacier-specific model calibration.

In this study the authors demonstrate that the glacier-specific and regional mass balance tuning procedures produce similar glacier projections at the regional scale, but that they (can) deviate significantly at the scale of individual glaciers. The agreement at the regional scale provides some confidence in sea level rise projections; the disagreement at small scales motivates further work aimed at refining global glacier models in order to better understand evolving water resources and natural hazards, which are more local in nature.

In addition, the paper is also one of the first to use the CMIP6 scenarios, and therefore complements a recent paper by Rounce et al.

This work is somewhat tangential to my main research interests. With that in mind, I found it a little difficult to wrap my head around the various global glacier models that are under development and considered in this paper (i.e., GloGEM, OGGM, and PyGEM). I think the paper could benefit from a clearer description of how the models differ from each other. The models are described in the text, but some of the description just points to previous publications. I'm not sure that lengthier descriptions would be necessary. Perhaps a table that lays out how the models handle mass balance and ice dynamics, while also including some of the strengths and weaknesses of each?

Other than that I felt that the paper was fairly easy to read and will make an important contribution to global glacier modeling.

**Specific comments**
L 33–36: I assume that these results come from using the glacier-specific mass balance observations? It is a little ambiguous because the previous sentences discuss regional vs. glacier-specific observations.

L 280: Should this be "21$^{\text{st}}$ century"?

L 288: What is meant by "similar state independent"? It seems that something is missing from this sentence.

L 291: I see the reference to Huss and Hock, but I also think that a brief description of the method/definition of discharge is warranted here. If I understand correctly, the discharge includes precipitation plus melt over the initial glacier area (i.e., the watershed area is fixed and the discharge includes the sum of glacier runoff and nonglacier runoff). The calculation does not take into account changes in evapotranspiration over the catchment, which could be significant, especially for glaciers that experience substantial retreat through the course of the simulations, which I think should be stated.

L 331–334: Another way to say this is that the large glaciers have long response times and therefore they are farther out of equilibrium with climate. For that reason, in a warming climate I would expect to see large glaciers tending to be in the lower left corners of Figures 5a,b.

L 385: Suggest deleting "logically".

L 462: Didn't you already state previously that Rounce et al., 2023 would be referred to as PyGEM?

L 468: There is a typo here or maybe a missing sentence? "...PyGEM. Table 1Figure 7A noteworthy distinction..."

---

## Author Comment (AC1)

**Reviewer 1**

**General comment**

**[RC1.01]** This is an interesting and well-designed paper by Zekollari et al, which provides a comparison of global glacier volume change under CMIP6. Additionally, the paper investigates the use of glacier-specific calibration techniques. Overall, the paper well written with the results thoroughly discussed. I have no major comments, but think the clarity of the text could be improved in certain places and have listed some minor comments below (including some adjustments to figures).

We thank the reviewer for their positive comment and general appreciation of the manuscript. We will improve the clarity of the manuscript by following the provided suggestions.

**Specific and technical comments**

**[RC1.02]** L90 – I find the description of this methodology a bit confusing, please consider rephrasing

This sentence will be rephrased to:
*The advantage of this approach is that the obtained mass balance parameters have (physically) realistic values, which fall within the literature ranges*

**[RC1.03]** L136 – Please state the year which the ice thickness dataset represents

The ice thickness is at the Randolph Glacier Inventory (RGI) date, which varies among (and also within) regions. This information will be added in the updated manuscript:
*In GloGEM, the ice thickness is from the consensus estimate of Farinotti et al. (2019a)* **at the RGI inventory date**, *which is deduced from the surface elevation (as provided in Farinotti et al., 2019a) to reconstruct the bedrock elevation*

**[RC1.04]** L168 – Please specify when the past climate starts (e.g. XXXX until 2020)

This information will be added:
*…(1980 until 2020)…*

**[RC1.05]** L171 – Please specify when the future climate ends

This will be included in the updated manuscript:
*(from 2020 until 2100)*

**[RC1.06]** L180 – What is the debiasing procedure?

The debiasing procedure allows for a consistency between the past climate data (ERA5) and the future climate model data over the common time period. This consistency occurs through a procedure that is described in the original GloGEM study (Huss and Hock, 2015). This will be further clarified in the updated manuscript:
*To ensure consistency between the observational/past (ERA5) and the future climate model data, a debiasing procedure is applied over the common 2000-2019 time period following the procedure described in Huss and Hock (2015)*

**[RC1.07]** L192 – I think 'Alternatively' should be changed to 'Additionally'

This should indeed be 'Additionally', which we will change in the updating the manuscript:
*Additionally, we also evaluate…*

**[RC1.08]** L227 – The sentence beginning 'In general, if…' is phrased in a slightly confusing way and could be reworked for better clarity – maybe something like 'In general, the mass balance calibration parameters are calibrated to give a more negative SMB for glaciers which have a lower mass balance than the regional average. This translates…'

This sentence was indeed a bit confusing. We will reformulate it along the lines you suggested, but will keep it as a single sentence to allow to make the contrast between this case (mass balance glacier < regional one) and the opposite case (mass balance glacier >

regional one, which we refer to as *and vice versa for higher mass balance*). The update sentence will be:

*In general, for glaciers with a mass balance lower than the regional one, the mass balance model parameters are calibrated to produce a more negative present-day mass balance, which translates into a more negative future mass balance and thus more substantial projected ice loss, and vice versa for higher mass balance*

**[RC1.09]** L284 – I would change the sentence 'For most other regions, these differences are even more outspoken..'. I do not think outspoken is a good choice of word, and this sentence is a bit dense and confusing to read. Could be changed to something like 'For most other regions, there is an even larger proportion of glaciers which show differences of more than 10% in their 2015-2050 volume projections. For instance, in High mountain Asia…'

We thank the reviewer for this suggestion and will change the text along these lines to:

*For most other regions, there is an even larger proportion of glaciers which show large differences in volume projections. For instance, in High-Mountain Asia (RGI regions 13, 14, 15), between 35-55% of all glaciers (with volume >0.1km³) have differences in the 2015-2050 volume projections of more than 10% depending on the calibration approach (Table S 3).*

**[RC1.10]** L288 – This could also be rephrased/ broken up into two sentences. Else, add '..independent of calibration methodology' or similar.

This will be updated following the second suggestion to:

*When considering the 2015-2100 volume evolution, the differences resulting from the calibration approaches are generally smaller, since a lot of the regions lose a large part of their mass, evolving to a similar (almost ice-free) state independent of the calibration methodology (Table S 3)*

**[RC1.11]** L299 – I do not understand what you mean by 'since discharge is calculated over initial glacier area'

The discharge values are calculated over the initial glacier area: i.e., even if the glacier becomes smaller (e.g., evolving from $10km^2$ to $5km^2$), the discharge is calculated over the initial area ($10 \ km^2$ in this example, accounting for the precipitation and snow melt over the deglaciated area). In the updated manuscript this will be explained by providing the following additional information:

*since discharge is calculated over initial glacier area, differences in precipitation result in differences in discharge*

Additionally, we will also provide additional information at the beginning of the paragraph when introducing the methodology to calculate discharge:

*…calculated over the initial glacier area, i.e. with fixed watershed area, following the method presented in Huss and Hock (2018), accounting for glacier and non-glacier runoff*

**[RC1.12]** Fig.2 – Please make the text in the 2015-2100 volume change boxes larger

In the updated figure, we will increase the font size of the text in the 2015-2100 volume change boxes

**[RC1.13]** Fig. 3 – Please also make the text large in this figure (for the calibrated values)

The text for the calibrated values has been made larger.

**[RC1.14]** Figs 2 and 3: It could be beneficial to move some of the panels to the supplementary material, so that the remaining panels can be more easily seen – as of now, they are quite busy.

Whereas these figures contain quite a lot of information, we would like to keep the same number of panels to clearly reflect the contrasts in the effect of the calibration for different types of glaciers in the main text. By increasing the font size of the figures (see previous comments), we have increased the readability of the figure.

**[RC1.15]** Fig.7 – If possible, please extend the width so that it takes up the full page width/ has the same dimensions as Fig.6

We will extend the width of both figures so that they take up the full page width.

**[RC1.16]** L424 – The explanation of why the inclusion of frontal ablation in GloGEM leads to less global volume loss does makes sense, but I had to read it a few times to understand. It would be good to add in a bit more detail here to make it extra clear.

In the new manuscript we will update this to contain additional detail:

*In GloGEM, since frontal ablation contributes to the total mass balance, a higher surface mass balance is needed to result in the same total mass balance as for the case without frontal ablation. As a consequence, if frontal ablation decreases (e.g. loss contact with ocean), the more positive mass balance dominates (vs. case without frontal ablation), resulting in less future ice loss. Given the very large uncertainties in modelled present-day and future frontal ablation, it is currently difficult to judge whether results from a setup with a relatively uncertain frontal ablation (GloGEM) or one in which it is not explicitly represented (OGGM setup used in this study) should be more trusted.*

**[RC1.17]** L468 – Formatting error (sentence beginning with 'Table1Figure7A')

Thank you for spotting this. This will be corrected in the new manuscript.

**[RC1.18]** Fig. 9 – It would be nice to move the GloGEM panel next to the panel which compares CMIP5 GloGEM to CMIP model ensemble, for ease of comparing the GloGEM results with each other. Also include a 'CMIP6' label under the GloGEM results like in the other panels. It could additionally be beneficial to add results showing the mean of the CMIP6 forced GloGEM/OGGM/PyGEM simulations, to be compared to the CMIP5 ensemble mean. In this case, you could move the CMIP5 GloGEM results to the GloGEM panel and then have an 'ensemble mean/ CMIP5 vs CMIP6' panel.

L519 – Here you compare CMIP5 ensemble to your new CMIP6 results, which I think would be strengthened through the above changes to Fig. 9. I understand the caveats to direct comparison that you explain in the text, but still think this would be a useful visual aid.

Adding GloGEM CMIP5 simulations to Figure 9 would be interesting to allow for a direct comparison on the effect of the CMIP5 vs. CMIP6 forcing for the GloGEM results. However, we do unfortunately not have the data at hand for doing so. More specifically, we did not re-run the latest version of GloGEM (which, among other novelties, now also includes a glacier-specific calibration) with CMIP5 simulations: the CMIP5 GloGEM simulations that we have at hand are those that we submitted at the time for GlacierMIP2, i.e. relying on an older version of the model (and without glacier-specific data at hand for calibration). As such, moving these (old) CMIP5 simulations (now the right-most column of the figure) to the GloGEM part of the figure would be misleading, as comparing this to the new CMIP6 simulations would be "comparing apple and oranges" (differences is volume can result from differences in climate forcing, model differences, and/or differences in model calibration).

For the two other models (OGGM and PyGEM), the model was re-run with exactly the same model version and calibration data for CMIP5 and CMIP6, therefore allowing for a comparison (and hence they are displayed next to one another). For the above reasons, we cannot aggregate the different model versions to create an ensemble mean for CMIP5 vs. CMIP6. We agree that this would have been an interesting comparison (if CMIP5 GloGEM simulations with latest version would have been available), but would like here to stick to the current organisation for the sake of clarity and to allow for 'clean' comparisons. We also believe that the current figure version accurately displays which data comes from which study: GloGEM (this study), OGGM v1.6.1 (this study), PyGEM (Rounce et al., 2023), and GlacierMIP2 (Marzeion et al., 2020; multi-model ensemble mean + global results from GloGEM version that was available at the time of that study). Following your suggestion, we will add the fact that GloGEM in this study is CMIP6 (i.e. add label below the bullet points), and also correct a typo for the CMIP5 GloGEM simulations in GlacierMIP2.

**[RC1.19]** L600 – This is a very long sentence that could benefit from being broken up into a few parts

This sentence will be split in two parts:

*In this respect, new glacier outlines (RGI v7.0, RGI Consortium, 2023) and other datasets with a (near) global coverage (e.g., on ice surface velocities and ice thickness reconstructions (e.g. Millan et al., 2022), and frontal ablation estimates (Kochtitzky et al., 2022)) will allow inverting glacier properties, calibrating model parameters, and evaluating model performance in a more advanced way. To combine this broad variety of datasets and observations, the field of large-scale glacier modelling will increasingly rely on machine learning and data assimilation (e.g. Bolibar et al., 2023; Cook et al., 2023; Jouvet and Cordonnier, 2023).*

**[RC1.20]** Fig. S3 – Make this figure take up the full page width; the panels are too small at the moment

This will be updated in the new manuscript.

---

## Author Comment (AC2)

**Reviewer 2 (Jason Amundson)**

**General comment**

> **[RC2.01]** This study investigates how mass balance data used for calibrating global glacier model projections affects model outcomes. Until recently, global glacier projections relied on either limited mass balance data from a small subset of glaciers or regional mass balance observations. Recent work has produced a data set of geodetic mass balances for all glaciers worldwide, thus enabling glacier-specific model calibration.
>
> In this study the authors demonstrate that the glacier-specific and regional mass balance tuning procedures produce similar glacier projections at the regional scale, but that they (can) deviate significantly at the scale of individual glaciers. The agreement at the regional scale provides some confidence in sea level rise projections; the disagreement at small scales motivates further work aimed at refining global glacier models in order to better understand evolving water resources and natural hazards, which are more local in nature.
>
> In addition, the paper is also one of the first to use the CMIP6 scenarios, and therefore complements a recent paper by Rounce et al. This work is somewhat tangential to my main research interests. With that in mind, I found it a little difficult to wrap my head around the various global glacier models that are under development and considered in this paper (i.e., GloGEM, OGGM, and PyGEM). I think the paper could benefit from a clearer description of how the models differ from each other. The models are described in the text, but some of the description just points to previous publications. I'm not sure that lengthier descriptions would be necessary. Perhaps a table that lays out how the models handle mass balance and ice dynamics, while also including some of the strengths and weaknesses of each?
>
> Other than that I felt that the paper was fairly easy to read and will make an important contribution to global glacier modeling.

We thank the reviewer for his positive review and are pleased to read that the manuscript was also easy to follow for a scientists that is not directly involved in the field of (large-scale) glacier evolution modelling. To allow for a better grasp of what the different large-scale glacier models do, what they have in common, and what is different, we will follow the suggestion by the reviewer and add a supplementary table (Table S2) that provides an overview.

**Specific comments**

> **[RC2.02]** L 33–36: I assume that these results come from using the glacier-specific mass balance observations? It is a little ambiguous because the previous sentences discuss regional vs. glacier-specific observations.

This is a good point and was indeed not very clear in the original abstract. We will now update the abstract by explicitly including the information that all these results are based on the mass balance calibration with glacier-specific observations:

*We project the 2015-2100 global glacier loss to vary between 25±15% (GloGEM) and 29±14% (OGGM) under SSP1-2.6 to 46±26% and 54±29% under SSP5-8.5 (ensemble median, with 95% confidence interval;* **calibration with glacier-specific observations***)*

> **[RC2.03]** L 280: Should this be "21st century"?

Yes, this should indeed have been '21st century'. We will update this to:

*…but here the calibration approach has an important effect on the 21st **century** transient evolution towards this deglaciation:…*

> **[RC2.04]** L 288: What is meant by "similar state independent"? It seems that something is missing from this sentence.

The sentence was unclear due to a missing comma and will be updated to:

*…since a lot of the regions lose a large part of their mass, evolving to a similar (almost ice-*

> *free) state*, *independent of the calibration methodology (Table S 3).*
> (i.e., it is not "state independent", but rather a "state", which is "independent of the calibtation methodology")
* * *
**[RC2.05]** L 291: I see the reference to Huss and Hock, but I also think that a brief description of the method/definition of discharge is warranted here. If I understand correctly, the discharge includes precipitation plus melt over the initial glacier area (i.e., the watershed area is fixed and the discharge includes the sum of glacier runoff and nonglacier runoff). The calculation does not take into account changes in evapotranspiration over the catchment, which could be significant, especially for glaciers that experience substantial retreat through the course of the simulations, which I think should be stated.

Yes, this is an accurate description, which we will incorporate when updating the manuscript as follows:

*…calculated over the initial glacier area, i.e. with fixed watershed area, following the method presented in Huss and Hock (2018), accounting for glacier and non-glacier runoff, but not including possible changes in evapotranspiration…*
* * *
**[RC2.06]** L 331–334: Another way to say this is that the large glaciers have long response times and therefore they are farther out of equilibrium with climate. For that reason, in a warming climate I would expect to see large glaciers tending to be in the lower left corners of Figures 5a,b.

Yes, this is true indeed. We will add this information in the updated sentence:

*…the little remaining regional ice volume is typically located in the largest glaciers (which have longer response times and thus take longer to disappear), which here tend to have a mass balance that is more negative than the regional one…*
* * *
**[RC2.07]** L 385: Suggest deleting "logically".

We will omit "logically" here.
* * *
**[RC2.08]** L 462: Didn't you already state previously that Rounce et al., 2023 would be referred to as PyGEM?

Yes, we did indeed already include this information earlier in the manuscript. The updated sentence will now be:

*Our modelled regional and global glacier volume changes agree well with those simulated with PyGEM by Rounce et al. (2023).*
* * *
**[RC2.09]** L 468: There is a typo here or maybe a missing sentence? "...PyGEM. Table 1Figure 7A noteworthy distinction..."

Thank you for spotting this. This will be corrected in the new manuscript.